# Integrated Methane Inversion (IMI) 2.0: an improved research and stakeholder tool for monitoring total methane emissions with high resolution worldwide using TROPOMI satellite observations

Lucas A. Estrada[1], Daniel J. Varon[1], Melissa Sulprizio[1], Hannah Nesser[2], Zichong Chen[1], Nicholas Balasus[1], Sarah E. Hancock[1], Megan He[1], James D. East[1], Todd A. Mooring[3], Alexander Oort Alonso[1], Joannes D. Maasakkers[4], Ilse Aben[4], Sabour Baray[5], Kevin W. Bowman[2], John R. Worden[2], Felipe J. Cardoso-Saldaña[6], Emily Reidy[6], Daniel J. Jacob[1]

[1]School of Engineering and Applied Science, Harvard University, Cambridge, Massachusetts, USA
[2]Jet Propulsion Laboratory, California Institute of Technology, Pasadena, California, USA
[3]Department of Earth and Planetary Sciences, Harvard University, Cambridge, MA, USA
[4]SRON Netherlands Institute for Space Research, Leiden, the Netherlands
[5]Environment and Climate Change Canada, Toronto, ON, Canada
[6]ExxonMobil Technology and Engineering Company, Spring, TX, USA

*Correspondence to*: Lucas A. Estrada (lestrada@g.harvard.edu)

**Abstract.** Satellite observations of atmospheric methane are a powerful resource for quantifying methane emissions over any region worldwide. The inverse methods needed to infer emissions from these observations require a high level of scientific and technical expertise as well as access to large computational and data processing resources. The Integrated Methane Inversion (IMI) is an open-access cloud computing tool designed for researchers and non-expert users to obtain total sector-resolved methane emissions worldwide at up to $0.25° \times 0.3125°$ ($\approx 25 \times 25$ km$^2$) resolution by analytical inversion of TROPOMI satellite observations with closed-form error characterization. Here we describe IMI version 2.0 with vastly expanded capabilities relative to the original version. Major developments include: (i) a new blended TROPOMI+GOSAT dataset for higher data quality, (ii) order-of-magnitude speedup in Jacobian matrix construction, (iii) improved error characterization through use of super-observations, (iv) improved methods for initial and boundary conditions, (v) adaptive spatial resolution linked to observational information content, (vi) incorporation of point source observations in state vector construction, (vii) option to optimize tropospheric OH (main methane sink), (viii) global inversion capability, (ix) Kalman filter option for continuous monitoring of emissions, (x) updated default prior emission inventories, (xi) option for lognormal error probability density functions to characterize emissions, (xii) additional output visualization (sectoral emissions, temporal variability), and (xiii) containerization to facilitate download to local computing facilities and operation as part of the US GHG Center. A 2023 annual inversion with 28-day temporal resolution for the contiguous US (CONUS) is presented as demonstration of IMI 2.0 capabilities.

# 1 Introduction

Methane, a powerful greenhouse gas, has become a top policy concern for mitigating anthropogenic climate change. Its short atmospheric lifetime (~9 years) and high warming effect (~82 times the global warming potential of $CO_2$ for a 20-year time horizon) make reducing methane emissions an attractive option for near-term climate benefits (Naik et al., 2013; Forster et al., 2021). Over 150 countries have joined the Global Methane pledge, committing to collectively reduce global methane emissions 30% by 2030 (Global Methane Pledge, 2023). Despite the recent attention from policymakers, global atmospheric concentrations of methane continue to rise rapidly, at a rate of 0.8% per year for 2020-2022 (Thoning et al., 2022). Quantification and monitoring of methane emissions from different sectors worldwide and with high resolution is crucial for understanding methane trends and developing mitigation methods to achieve policy goals.

Methane has anthropogenic sources from many sectors including livestock, oil and gas operations, coal mining, landfills, wastewater treatment, and rice cultivation (Saunois et al., 2020). Wetlands are the dominant natural source. Emissions are estimated using either bottom-up or top-down methods. Bottom-up methods either apply emission factors to units of activity or use process-based models, but emission factors can have large uncertainties, activity data may lag by several years, and process-based models are often highly parametrized. Top-down methods use observations of atmospheric methane combined with an atmospheric transport model to infer emissions, but there may be errors in the observations and in the modeling of transport. Inversions of atmospheric observations blend these two methods by using bottom-up estimates as prior information and adjusting them to optimize the fit to observations through the atmospheric transport model, with formal characterization of errors by Bayesian synthesis (Brasseur and Jacob, 2017). Results from these inversions can provide guidance for improving policy-relevant bottom-up inventories.

Satellite observations of atmospheric methane have greatly increased the potential of inverse analyses by providing global continuous coverage and high data density. Satellites retrieve methane dry column mixing ratios $X_{CH4}$ with sensitivity down to the surface by measuring backscattered solar radiation in the shortwave infrared (SWIR) (Jacob et al., 2016). Observations are available from a range of satellite instruments to quantify emissions from the global scale down to point sources (Jacob et al., 2022). The TROPOMI instrument launched in October 2017 has daily global coverage with $5.5 \times 7$ km$^2$ nadir pixel resolution (Veefkind et al., 2012; Lorente et al., 2021) and presently provides the most spatially dense top-down resource for global mapping of total methane emissions. A number of inverse studies have used TROPOMI observations to quantify methane emissions globally and for specific regions (Cusworth et al., 2020; Y. Zhang et al., 2020; McNorton et al., 2022; Chen et al., 2023; Li et al., 2023; Naus et al., 2023; Shen et al., 2023; Tsuruta et al., 2023; Varon et al., 2023; Yu et al., 2023; Nesser et al., 2024). Point source imagers including GHGSat, EMIT, PRISMA, and Sentinel-2 provide individual information on large point sources (Jacob et al., 2022).

Inversions of satellite data require a high level of technical and scientific expertise, as well as large computational and data processing resources. Transparency and accessibility of the methods are essential for making the resulting emission estimates actionable by stakeholders. These stakeholders may be from government agencies at all levels (municipal to national), international agencies, non-governmental organizations, industry, and advocacy groups. Greater impact can be achieved if the methods are usable by the stakeholders themselves. New inversion tools have recently been developed for this purpose including the Community Inversion Framework (Berchet et al., 2021) and version 1.0 of the Integrated Methane Inversion (IMI; Varon et al., 2022).

The IMI (https://carboninversion.com/) is specifically designed to enable researchers and non-expert stakeholders to exploit TROPOMI satellite data for optimizing total methane emission estimates at up to 25-km resolution. It uses a state-of-the-art analytical inversion method with closed-form error characterization documented in the research literature. It operates on the Amazon Web Services (AWS) cloud, where both TROPOMI data and the atmospheric transport model (GEOS-Chem) reside, thus avoiding the need for local computing resources and instead bringing compute to data. The IMI has a user-friendly interface to enable stakeholders to optimize emission estimates for any selected domain and period through a configuration file, with default or user-defined prior estimates and error specifications. It features an open-source codebase, comprehensive documentation (https://imi.readthedocs.io/en/latest/), and frequent versioning to keep the methods up to date with current research. The IMI is actively being used for research applications (Baray et al., 2023; Chen et al., 2023; Nathan et al., 2023; Varon et al., 2023; Hemati et al., 2024; Vara-Vela et al., 2024; Hancock et al., 2025).

Varon et al. (2022) documented the initial release of IMI 1.0, limited at the time to regional inverse analyses of TROPOMI observations with $0.25° \times 0.3125°$ ($\approx 25 \times 25$ km$^2$) or $0.5° \times 0.625°$ ($\approx 50 \times 50$ km$^2$) resolution. Since then, the IMI has undergone substantial development, and we document here the greatly enhanced capabilities of IMI 2.0 including the major new features listed in Table 1. These IMI 2.0 advancements improve the overall flexibility of the IMI for a wider range of scientific and stakeholder applications from regional to global scales and with low temporal latency to enable continuous monitoring. Sect. 2 gives a summary description of IMI 1.0 and Sect. 3 describes the major new features of IMI 2.0. A 1-year demo inversion for the contiguous US (CONUS) is presented in Sect. 4. Current limitations for future development are discussed in Sect. 5.

**Table 1: Integrated Methane Inversion (IMI) capabilities**

| IMI 1.0 (Varon et al., 2022) | IMI 2.0 New Capabilities [a] |
|---|---|
| • Emission optimization for regional domains with 25-50 km resolution by analytical inversion of TROPOMI observations <br><br> • Smoothed TROPOMI fields as boundary conditions <br><br> • Open-source code on AWS cloud with user-friendly interface <br><br> • IMI preview for visualizing data and assessing information content before performing inversion. <br><br> • Output data and imagery for posterior (optimized) fluxes with error statistics | 1. Blended TROPOMI+GOSAT dataset compatibility <br> 2. Jacobian matrix construction speed-up <br> 3. Super-observations <br> 4. Optimization of boundary conditions <br> 5. Adaptive information-based spatial resolution <br> 6. Point source incorporation in state vector construction <br> 7. Optimization of methane sink from OH <br> 8. Global inversion capability <br> 9. Low-latency emission updates (continuous monitoring) <br> 10. New bottom-up emission inventories as prior estimates <br> 11. Lognormal error statistics for prior emission estimates <br> 12. Enriched output information <br> 13. Docker container for code download to local systems |

[a] Numbers correspond to subsections in Sect. 3.

## 2 Integrated Methane Inversion (IMI) 1.0

We start with a summary description of the Integrated Methane Inversion (IMI) 1.0, previously described by Varon et al. (2022), to provide context for the new developments in IMI 2.0. IMI is designed as an open-source software tool for use on the cloud or local clusters to infer methane emissions from TROPOMI satellite observations using GEOS-Chem as the forward atmospheric transport model. The TROPOMI data are from the latest operational retrieval version archived on the AWS cloud (currently v02.06.00; Lorente et al., 2023). The data are filtered to remove retrievals with QA value ≤ 0.5, water pixels, and pixels south of 60°S. GEOS-Chem is a global 3-D chemical transport model (Wecht et al., 2014) that can operate in regional mode at up to 0.25°×0.3125° resolution using archived meteorological input from the NASA Goddard Earth Observing System - Fast Processing (GEOS-FP) and MERRA-2 datasets. The IMI uses GEOS-Chem Classic with shared-memory parallelization (Bey et al., 2001). Smoothed TROPOMI fields are used as GEOS-Chem boundary conditions to ensure consistency with the TROPOMI data within the inversion domain (Shen et al., 2021).

IMI 1.0 conducts methane inversions on the native 0.25°×0.3125° GEOS-FP grid (or alternatively the 0.5°×0.625° MERRA-2 grid) over the TROPOMI record from May 2018 to present. Users perform methane inversions by filling out a simple configuration text file (YAML format) selecting their region and period of interest. The region can be set as a rectilinear

domain (latitude and longitude boundaries) or by providing a shapefile with any geometry. The state vector optimized by the inversion consists of temporal mean emissions for the period of interest in individual emitting grid cells (including offshore emissions) within the region of interest, plus buffer clusters surrounding the region of interest to correct boundary conditions. Prior estimates for the inversion are from a default IMI library of bottom-up emission inventories, but users can substitute their own. The cloud compatibility of the IMI allows users to leverage the vast computation resources of the AWS cloud for performing inversions without need for a local compute cluster. It takes advantage of input data already being resident on the cloud including the TROPOMI observations, the smoothed TROPOMI fields used as boundary and initial conditions for the inversions, the GEOS-FP and MERRA-2 meteorological datasets used by GEOS-Chem, and the bottom-up emission inventories used as prior estimates. Advanced users can adjust the inversion settings via the configuration file or through manual update of the inversion workflow, which involves a collection of Bash™ and Python® scripts.

The IMI follows the established Bayesian analytical inversion technique described by Brasseur and Jacob (2017) to optimize an emission state vector $x$ (2-D gridded fluxes) by minimizing the cost function $J(x)$:

$$J(x) = (x - x_A)^T S_A^{-1}(x - x_A) + \gamma(y - Kx)^T S_o^{-1}(y - Kx) \qquad (1)$$

Here $x_A$ is the prior estimate, $y$ denotes the observations assembled into a vector, $S_A$ is the prior error covariance matrix, $S_o$ is the error covariance matrix of the observational system, $K$ is the Jacobian matrix describing the forward model sensitivity of the observations to perturbations in emissions, and $\gamma \in [0,1]$ is a regularization parameter to compensate for unaccounted error covariances in the observational system (Lu et al., 2021). The optimized emissions $\hat{x}$ (posterior estimate) are obtained by solving analytically for the cost function minimum where $\partial J/\partial x = 0$:

$$\hat{x} = x_A + (\gamma K^T S_o^{-1} K + S_A^{-1})^{-1} \gamma K^T S_o^{-1}(y - Kx_A) \qquad (2)$$

The relationship of emissions to concentrations is linear so that $K$ fully defines the forward model for the purpose of the inversion. It is constructed column by column by running embarrassingly parallel perturbation simulations with the GEOS-Chem forward model.

A major advantage of the analytical solution is that it provides a closed-form expression for the posterior error covariance matrix $\hat{S}$:

$$\hat{S} = (\gamma K^T S_o^{-1} K + S_A^{-1})^{-1} \qquad (3)$$

$\hat{\mathbf{S}}$ characterizes the error on $\hat{x}$, and comparison to $\mathbf{S_A}$ quantifies the information content from the inversion. This is critically important for satellite observations, which generally do not fully constrain the state vector. From $\hat{\mathbf{S}}$ we derive the averaging kernel matrix for the inversion $\mathbf{A} = \partial \hat{x} / \partial x = \mathbf{I} - \hat{\mathbf{S}} \mathbf{S_A^{-1}}$, which measures the sensitivity of the inverse solution to the true state. The trace of $\mathbf{A}$ measures the degrees of freedom for signal (DOFS), representing the number of independent pieces of information on the state vector obtained from the inversion, and the diagonal elements $a_{ii} = \partial \hat{x}_i / \partial x_i \in [0,1]$ measure the ability of the inversion to quantify state vector element $x_i$ independently from the prior estimate (fully if $a_{ii} = 1$, not at all if $a_{ii} = 0$). Another advantage of the analytical solution is that once $\mathbf{K}$ has been constructed, any ensemble of analytical inversions exploring the sensitivity to different inversion parameters can be easily and rapidly generated (Chen et al., 2022).

The regularization parameter $\gamma$ is designed to avoid overfit to the observations. Users can choose an optimal value of $\gamma$ such that $(\hat{x} - x_A)^T \mathbf{S_A^{-1}} (\hat{x} - x_A) \approx n \pm \sqrt{2n}$ following Lu et al. (2021), corresponding to the expected value of the chi-square distribution. Again, once $\mathbf{K}$ has been constructed, it is easy to conduct inversions with different values of $\gamma$ in order to determine the optimal value.

Regional inversions require unbiased boundary conditions, as biases in boundary conditions would propagate to the optimized emissions in the inversion domain. The IMI maintains a global 3-D archive of bias-corrected GEOS-Chem fields (called smoothed TROPOMI fields) to serve as unbiased boundary conditions for any TROPOMI inversion domain or period. The archive is produced by correcting a global continuous GEOS-Chem simulation at 4°×5° resolution with smoothed TROPOMI concentrations (12°×15° spatially and +/- 15 days temporally) and applying zonal mean corrections over the oceans. In IMI 1.0, boundary conditions are further corrected in the inversion using buffer grid cell clusters surrounding the region of interest (Shen et al., 2021).

The IMI includes a preview feature to quickly estimate the information content and computational cost of a proposed inversion before investing resources in running the full inversion. The preview provides maps of mean TROPOMI concentrations, observation density, prior emissions, estimated averaging kernel sensitivities, and SWIR albedo for the selected inversion period and domain, along with a cost estimate for running on the cloud. The averaging kernel sensitivities $a_{ii}$ and DOFS $= \sum_i a_{ii}$ are estimated without doing the actual inversion by assuming uniform observations and a simple transport parameterization:

$$a_{ii} = \frac{\sigma_{a_i}^2}{\sigma_{a_i}^2 + \frac{(\sigma_o/k)^2}{m/n}}, \qquad (4)$$

where $\sigma_{a,i}$ is the prior error standard deviation for $x_{a,i}$, $\sigma_o$ is the observational error standard deviation, $m$ is the number of satellite observations in the domain, $n$ is the number of state vector elements, and $k$ relates $X_{CH4}$ to local emissions with a

simple advection-diffusion formulation (Nesser et al., 2021). The DOFS from the preview estimate how well an inversion with the specified configuration will be able to quantify emissions, allowing users to modify their inversion domain and/or period to improve this ability.

The IMI has extensive documentation available through its website (https://imi.seas.harvard.edu). The source code has undergone incremental development since 1.0 with official releases of IMI 1.1 and 1.2 and has an international following of users. Version 2.0 described here represents a transformational leap in the capabilities of the IMI.

## 3 New features in IMI 2.0

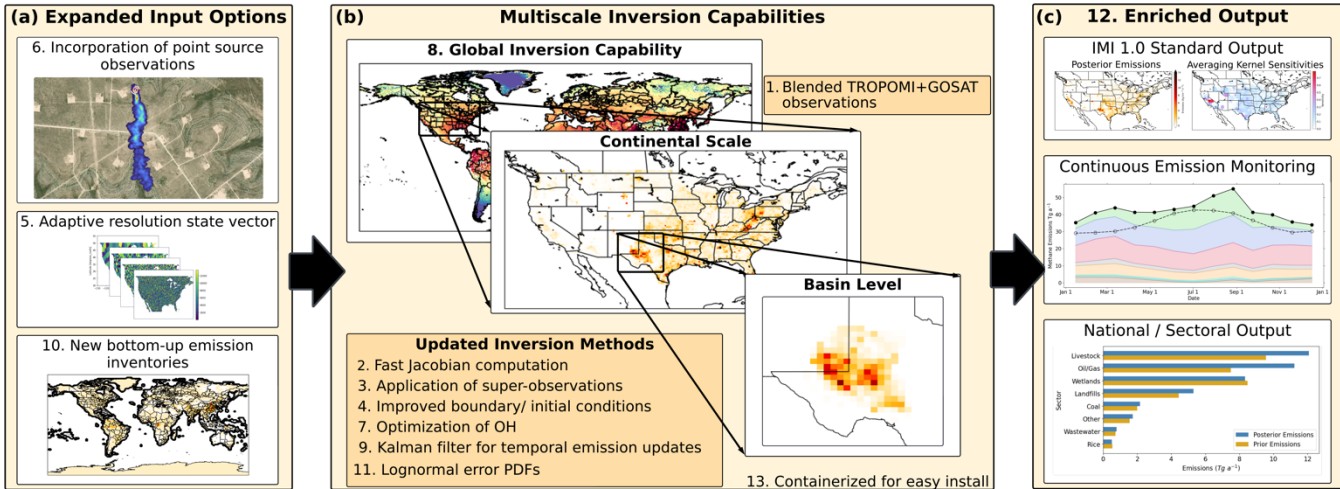

**Figure 1: Summary of major new features in IMI 2.0 divided into three categories: (a) expanded input options, (b) multiscale inversion capabilities, and (c) enriched output. Numbering of individual features corresponds to the relevant subsection in Sect. 3. Methane plume map in the top left panel is from Carbon Mapper (Methane Dashboard | Carbon Mapper, 2024).**

IMI 2.0 improves IMI 1.0 on a number of fronts including accuracy, performance, capability, and versatility for scientific and stakeholder applications. Table 1 lists the principal features, and these are illustrated in Fig. 1. The following subsections elaborate on each feature in the numerical order of the Table. IMI 2.0 gives users the option of a new blended TROPOMI+GOSAT dataset removing most artifacts from the operational product (Sect. 3.1). Computational performance is improved several-fold through fast construction of the Jacobian matrix (Sect. 3.2) and through use of super-observations that also better account for observational error correlation (Sect. 3.3). Boundary/initial conditions are improved, including synchronous representation of the stratosphere, and allowing for boundary condition optimization through the inversion (Sect. 3.4). The capability to conduct inversions over large domains is enhanced by an adaptive k-means clustering scheme (Sect. 3.5) that accounts for information on large point sources (Sect. 3.6). The methane sink from oxidation by tropospheric

OH can be optimized (Sect. 3.7). Inversions can be conducted on the global scale (Sect. 3.8). IMI 2.0 features a new capability for time-dependent inversions, allowing continuous monitoring of the evolution of emissions over a selected region of interest (Sect. 3.9). New prior emission inventories are introduced (Sect. 3.10). Users can select lognormal error
probability density functions (PDFs) for prior emissions to resolve the heavy tail in the distribution (Sect. 3.11). The inversion output includes new national and sectoral information on posterior emission estimates as well as time series visualizations (Sect. 3.12). The IMI is now in a software container to facilitate downloads to local computing clusters (Sect. 3.13). Most improvements in IMI 2.0 are the heritage of research studies referenced in the appropriate subsections.

Table 2 gives a summary list of the user-controlled variables in the IMI 2.0 configuration file. Full documentation is at https://imi.readthedocs.io. The default settings enable a basic user to run the IMI as simply as specifying a rectilinear inversion domain and time period. More advanced users have a wide range of options. The last column lists the settings used in the US demo inversion of Sect. 4.

**Table 2: IMI 2.0 configuration variables [a]**

| Setting | Default | Options | US demo (Sect. 4) |
|---|---|---|---|
| Spatial domain | Regional rectilinear [b] | Shapefile or global | CONUS shapefile |
| Time period | Start and end dates | Continuous monitoring | Jan 1 – Dec 31, 2023 |
| Spatial resolution | 0.25°×0.3125° | 0.5°×0.625°, 2°×2.5°, 4°×5° | 0.25°×0.3125° |
| Observation dataset | TROPOMI | Blended TROPOMI+GOSAT | Blended TROPOMI+GOSAT |
| Over water observations | No | Yes | No |
| Point source datasets [c] | SRON | Carbon Mapper, IMEO | SRON |
| Temporal updates (Kalman filter) | No | Yes [d] | Yes |
| State vector dimension | Native resolution | Clustering [e] | Clustering [f] |
| Country tag [g] | No | Yes | Yes |
| Jacobian construction grouping [h] | 5 | Any positive number | 5 |
| Boundary condition optimization [i] | Yes | No | Yes |
| OH optimization [j] | No | Yes | No |
| Prior emission inventories | Table 3 | Other inventories | Table 3 |
| Prior error PDFs [k] | Normal | Lognormal | Normal |
| Prior error standard deviation | 0.5 [l] | Any positive value | 0.5 |
| Observational error, ppb [m] | 15 | Any positive value | 15 |
| Regularization parameter $\gamma$ | 1.0 | Any positive value | 0.2 |

[a] Full documentation is at https://imi.readthedocs.io.
[b] Defined by latitude/longitude boundaries.

 *c* To impose native resolution in the inversion for locations of detected large point sources. Includes option to only include point sources above a certain value (default set to 2500 kg h$^{-1}$) and above a threshold of repeated observations above that value (default 50). The default values are used in the CONUS demo.

*d* With choices for the update frequency (e.g. 1-week/ 1-month) and specification of the nudge factor, which is the percent weight of the original prior emissions to apply to the prior estimate for the next inversion interval. See Sect. 3.9 and Varon et al. (2023) for more details.

*e* k-means information-based clustering to reduce state vector size with default or user-set parameters and including adaptive capability in Kalman filter mode (Sect. 3.5). Users can adjust cluster size distribution by setting the maximum cluster size (default 64 grid cells) and a threshold for information content per cluster (default of DOFS/$n$ from the IMI preview).

*f* With specification of 600 state vector elements.

*g* Used in k-means clustering algorithm to contain individual clusters within national borders, and also for national output of emissions by sectors.

*h* Number of state vector elements perturbed (columns of Jacobian matrix constructed) in each GEOS-Chem simulation, chosen to balance CPU time and wall time (Sect. 3.2).

*i* Setting ignored in global inversions. Error standard deviation configured by user with default of 10 ppb.

*j* Optimization of domain mean (regional inversions) or hemispheric mean (global inversions) tropospheric OH concentrations expressed as methane loss frequencies and with user-adjustable default error standard deviation of 10%.

*k* For prior emission estimates.

*l* Fractional error standard deviation for a normal error PDF, geometric standard deviation for a lognormal error PDF.

*m* Error standard deviation for individual observations. including contributions from measurement, retrieval, representation, and forward model errors.

## 3.1 Blended TROPOMI+GOSAT dataset

TROPOMI observations are available from May 2018 to present. The current TROPOMI operational retrieval is posted with 2-3 day latency on the AWS cloud (presently version 02.06.00; Sentinel-5P Level 2 - Registry of Open Data on AWS, 2024; Lorente et al., 2023) and is accessed there by the IMI as default. The retrieval is regularly updated to resolve artifacts from surface albedo, aerosols, clouds, and cross-track detector differences (Lorente et al., 2023). We provide an option to use the blended TROPOMI+GOSAT dataset of Balasus et al. (2023), which applies machine learning to correct the TROPOMI version 02.06.00 retrieval with the more accurate but much sparser retrieval from the GOSAT satellite instrument (Parker et al., 2020). Glint observations over water were previously filtered out due to high biases and artifacts in the TROPOMI retrieval, but with the higher fidelity of the blended TROPOMI+GOSAT dataset we include an option to include these observations in the inversion. This can help to quantify offshore emissions, adding to the information from when the offshore emission plume is transported over land. The blended TROPOMI+GOSAT data are available on the AWS cloud for the full duration of the TROPOMI record and we keep them current for use in the IMI. Comparison of the blended TROPOMI+GOSAT and operational TROPOMI datasets through the IMI preview can be insightful for identifying retrieval artifacts.

## 3.2 Fast Jacobian construction

Construction of the Jacobian matrix, **K**, is the most computationally expensive component of the IMI and has previously limited the size of the state vector to ~2,000 elements. **K** is constructed column by column by conducting GEOS-Chem simulations over the inversion time period perturbing individual state vector elements and collecting the resulting changes in concentrations. This was done in IMI 1.0 with separate GEOS-Chem simulations for each state vector element, and additional overhead was incurred by compiling methane emissions in GEOS-Chem with the Harmonized Emission Component (HEMCO; Lin et al., 2021). For IMI 2.0 we made several improvements to our Jacobian construction practices.

To avoid the effect of small nonlinearities in the advection code (Lin and Rood, 1996), we construct our Jacobian columns by specifying a low methane background and initial conditions (1 ppb), applying a high relative perturbation such that the median emission is $10^{-8}$ kg m$^{-2}$ s$^{-1}$ for each grid cell in the perturbed state vector elements, and setting all other emission state vector elements to zero. To improve computational performance, we modified GEOS-Chem to represent methane emissions from multiple state vector elements in a single simulation as independently transported tracers, so that several columns of the Jacobian can be constructed at once with no additional overhead. Additionally, we apply HEMCO to precompute total emissions for individual grid cells before running GEOS-Chem, reducing the time to read in emissions. Running GEOS-Chem with a large number of tracers can increase wall time, because GEOS-Chem Classic simulations are limited to a single node (shared-memory parallelization), while multiple GEOS-Chem simulations for Jacobian construction can be spread across nodes. We optimized the number of tracers per GEOS-Chem simulation to balance the total CPU time (which decreases with the number of tracers) and the wall time (which increases with the number of tracers, because the IMI then uses fewer compute nodes). Tests on the Harvard supercomputing cluster using 32 CPUs per GEOS-Chem simulation indicate an optimum of 5 tracers per simulation (Fig. 2). This yields a 5-fold speed-up in the construction of the Jacobian matrix relative to single-tracer simulations in total compute hours, traded against a 60% increase in wall time, as shown in Fig. 2. Precompiling emissions with HEMCO yields an additional 2-fold speed-up, for an overall 10-fold decrease in CPU cost and a net decrease in wall time. Users with a large number of available nodes can reduce wall time at the expense of CPU time by choosing fewer tracers per simulation as specified in the configuration file (Table 2).

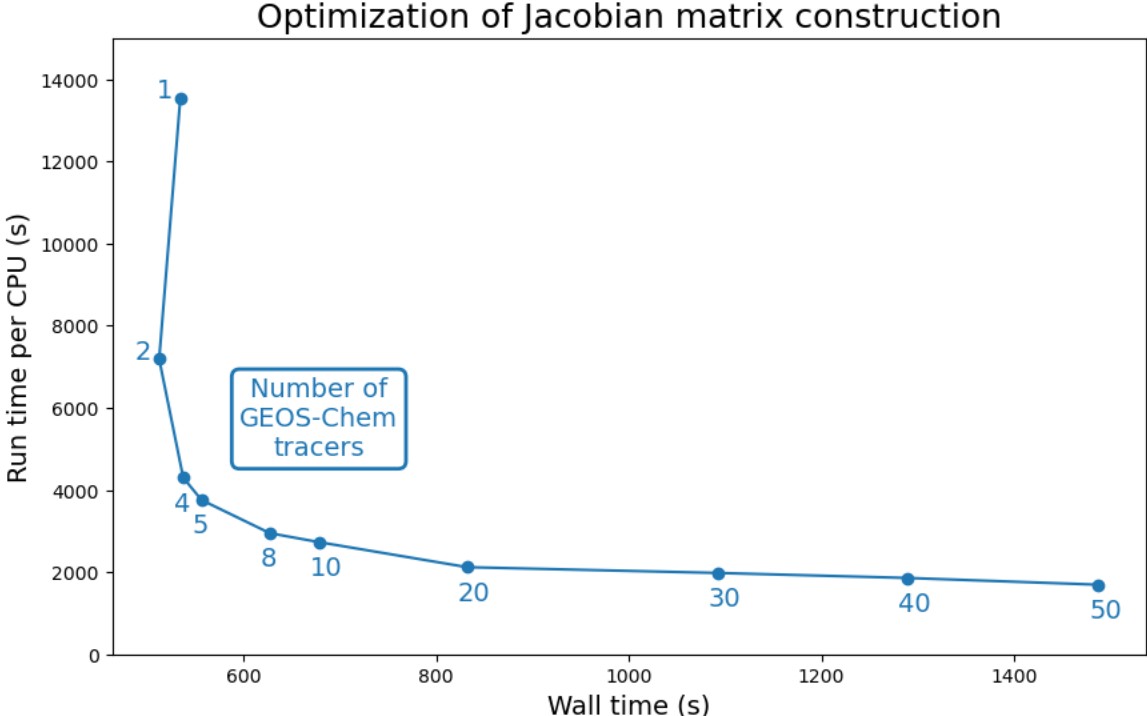

**Figure 2: Performance optimization of the number of state vector elements (transported methane tracers) in GEOS-Chem simulations used to construct the Jacobian matrix for the IMI. The plot shows the run time per CPU and the total wall time for GEOS-Chem simulations conducted with different numbers of transported tracers. A simulation with 5 tracers (elbow of the curve) provides the optimal performance for CPU and wall time. Tests were done on the Harvard supercomputer cluster for 1-month GEOS-Chem simulations with 32 CPUs (Intel 8480+ processors) at 0.25°×0.3125° resolution.**

### 3.3 Super-observations

IMI 2.0 uses super-observations as the TROPOMI observation vector $y$ in the inversion, following the work of Chen et al. (2023). Super-observations average all individual TROPOMI observations for a given GEOS-Chem grid cell and TROPOMI orbit. We call them super-observations following Eskes et al. (2003) because they have lower error than individual observations. Loss of information in this averaging of individual observations is negligible because GEOS-Chem model values are the same for all observations being averaged, and retrieval averaging kernels are similar. Using super-observations reduces the storage size and computational time associated with performing the inversion step of the IMI by reducing the dimension of $y$. Additionally, it provides a better characterization of observational error correlations to avoid overfit in the inversion. Using individual observations in the inversion with a diagonal observational error correlation matrix assumes that errors in individual observations are uncorrelated. In fact, transport errors for individual TROPOMI observations within a GEOS-Chem grid cell are perfectly correlated, and retrieval errors may be correlated as well. Using the residual error method (Heald et al., 2004), Chen et al. (2023) derived the observational error variance $\sigma_{super}^2$ for a super-observation averaging $P$ individual TROPOMI observations:

$$\sigma_{super}^2 = \sigma_{retrieval}^2 \left( \frac{1 - r_{retrieval}}{P} + r_{retrieval} \right) + \sigma_{transport}^2 \qquad (5)$$

where $\sigma_{transport}^2$ is the error variance associated with GEOS-Chem transport, $\sigma_{retrieval}^2$ is the single retrieval error variance, and $r_{retrieval}$ is the error correlation coefficient for the $P$ observations being averaged. We use $r_{retrieval} = 0.55$ and $\sigma_{transport} = 4.5$ ppb following Chen et al. (2023) for an inversion at 0.25°×0.3125° resolution. $\sigma_{retrieval}$ is set by the user, with a default value of 15 ppb from previous applications of the residual error method to TROPOMI data (Qu et al., 2021; Shen et al., 2021; Chen et al., 2023). The improved observational error characterization from using super-observations reduces our reliance on $\gamma$ to account for overfit, allowing $\gamma$ to be set closer to 1.

### 3.4 Smoothed TROPOMI fields as boundary and initial conditions

It is critical for the IMI to use unbiased initial and boundary conditions (IC/BCs), because bias will otherwise propagate to the emission correction. IMI 1.0 used an archive of spatially and temporally smoothed TROPOMI fields as IC/BCs to avoid systematic bias, and further optimized buffer clusters around the region of interest to correct for BC errors (Sect. 2). In IMI 2.0 we make four updates to the treatment of IC/BCs. First, we improve the process for generating smoothed TROPOMI fields to include a more accurate and synchronous stratosphere (Mooring et al., 2024). Second, we produce an additional

parallel archive of smoothed fields for the blended TROPOMI+GOSAT product (Balasus et al., 2023). Third, we allow for the optimization of BCs as well as buffer clusters. Fourth, we conduct the smoothing backward in time (rather than centered in time) to allow for near-real-time (low-latency) applications. The archives of smoothed TROPOMI fields for both the operational product and the blended product are kept current with ~1 month latency.

To produce the smoothed TROPOMI archives, we start from a global GEOS-Chem simulation (version 14.3.1) at horizontal resolution of $2° \times 2.5°$, improving upon the $4° \times 5°$ resolution of IMI 1.0, with 47 vertical layers extending to 0.01 hPa. The simulation uses the default prior emission estimates for the IMI (Table 3). It starts on April 1, 2018 (one month before the start of the TROPOMI record) with ICs from a separate GEOS-Chem simulation initialized in 1985 (Mooring et al., 2024). This multidecadal spin-up simulation uses monthly interpolated global surface observations of methane concentrations from the NOAA GLOBALVIEW flask dataset as BCs (Murray, 2016). GEOS-Chem transports these surface BCs throughout the atmosphere, a process that takes years for the stratosphere (Chabrillat et al., 2018). In this manner, we create an April 1, 2018 IC that is consistent with both long-term trends in tropospheric methane and stratospheric transport (Mooring et al., 2024).

The 3-D methane concentration fields from our emissions-driven global simulation starting on April 1, 2018 from this unbiased synchronized atmosphere are archived every 3h from 1 April 2018 until present. We correct these fields to the TROPOMI satellite observations to generate the smoothed TROPOMI fields. For each observation, we apply the TROPOMI operator, which describes the sensitivity of the observation to different vertical levels, to the co-located GEOS-Chem vertical profile, giving us pairs of TROPOMI $X_{CH4}$ and simulated GEOS-Chem $X_{CH4}$. We average these $X_{CH4}$ values on a $2° \times 2.5°$ daily grid and subtract them to get a GEOS-Chem column bias $\Delta X_{CH4} = X_{CH4, \text{GEOS-Chem}} - X_{CH4, \text{TROPOMI}}$. The column-bias fields are then smoothed with a rolling average spatially to $10° \times 12.5°$ and temporally to 15 days back in time. If there are no observations for the past 15 days, we extend the temporal smoothing to 30 days. For grid cells with no column-bias information after smoothing (such as over open oceans or high latitudes in the winter), we use a zonal average of the column bias for that latitude band or for the closest latitude band where information is available. The resulting $2° \times 2.5°$ daily smoothed column-bias fields are removed from each of the 47 layers to yield the bias-corrected 3-D GEOS-Chem fields (called smoothed TROPOMI fields) used as IC/BCs in the inversions.

Although our smoothed TROPOMI fields are intended to avoid systematic IC/BC bias in inversions of TROPOMI data, there may still be error in the BCs not captured by the smoothing, particularly in areas with few observations (oceans, high latitudes). In IMI 1.0 this was corrected by optimizing emissions in buffer clusters surrounding the region of interest. In IMI 2.0 we implement a new option to allow optimization of the BCs as part of the inversion. When enabled, BC optimization adds four additional elements to the state vector, one for each edge of the GEOS-Chem domain (when using a custom shapefile the domain is padded to be rectilinear). Each edge is then optimized as a constant correction as part of the

inversion. Optimization of BCs can be done in place of or in addition to optimization of buffer clusters. Recent work by Nesser et al. (2024b) finds that optimization of BCs is preferable to optimization of buffer clusters and has the advantage of being more physically based.

## 3.5 Adaptive state vector clustering

The maximum resolution of the IMI is set by the native 0.25°×0.3125° grid cell resolution of the GEOS-Chem forward model. Optimizing emissions at such a high resolution may be computationally burdensome for large domains and may not be justified by the information content of the observations. We introduce in IMI 2.0 an adaptive k-means clustering algorithm following Nesser et al. (2021) that clusters individual grid cells on the basis of proximity and information content. This maintains high resolution in areas of high emissions and dense observations while smoothing the solution in areas with

weak emissions or insufficient observations. The algorithm is adaptive in determining the best clustering to apply given user specification of a desired number of state vector elements and minimum resolution, and also in adjusting to changing observing conditions for temporally resolved inversions (Sect. 3.9). It is interactive with the user through the IMI preview. Users have options to exclude selected grid cells from the clustering (force them to remain at native resolution), on the basis of, for example, ancillary observations of large point sources (Sect. 3.6). To improve national emission accounting, users

also have the option to ensure that clusters respect country boundaries.

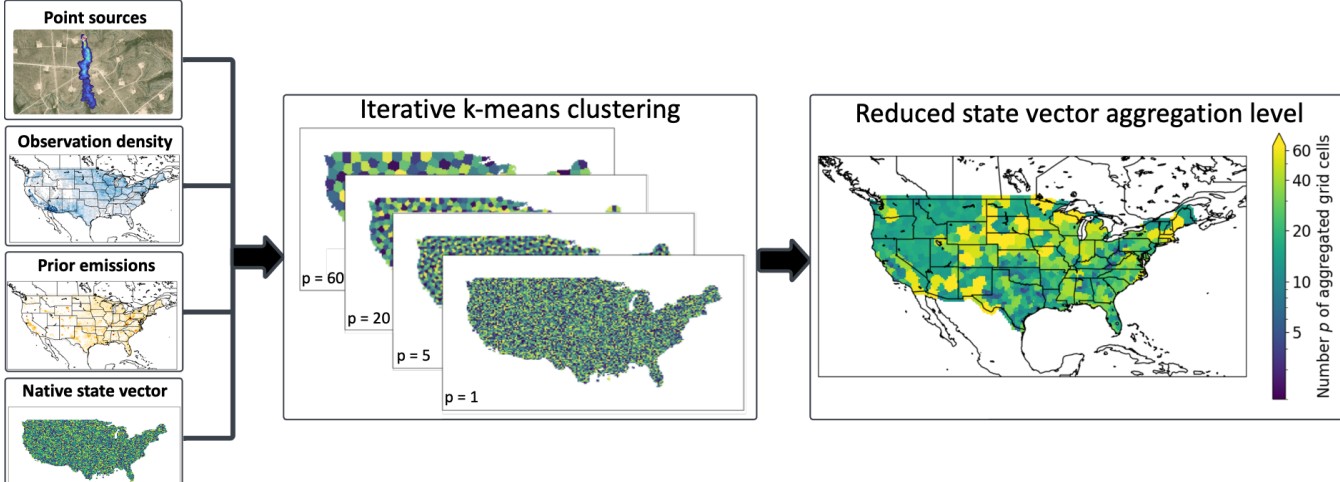

**Figure 3: Spatial clustering of state vector elements in IMI 2.0 using iterative k-means and illustrated for the CONUS emission state vector in the demo of Sect. 4. The native-resolution state vector, consisting here of 0.25°×0.3125° grid**

**cells, is transformed into an ensemble of multi-sized state vector elements aggregating $p$ grid cells depending on information content. The algorithm considers point source information, observation density, prior emissions, and country borders to construct an optimally reduced state vector (right panel) based on the user-specified number of state vector elements, maximum cluster size, and information content per element. The reduced state vector in the right panel has 600 elements with the number $p$ of 0.25°×0.3125° grid cells being aggregated ranging from $p = 1$ (no**

 **aggregation) to $p \approx 64$. Methane plume map in the top left panel is from Carbon Mapper (Methane Dashboard | Carbon Mapper, 2024).**

Figure 3 illustrates our clustering algorithm with the example of the CONUS demo in Sect. 4. The user configures their desired number of state vector elements and maximum cluster size ($2° \times 2.5°$ by default). The clustering algorithm first applies the IMI preview to estimate averaging kernel sensitivities, $a_{ii}$, (Eq. (4)), for each native resolution grid cell in the domain on the basis of the number of super-observations and the prior emission estimates. Cells that have an $a_{ii}$ greater than a clustering threshold are kept at native resolution ($p = 1$) in the state vector. They may also be kept at native resolution if flagged by the user or by a point source dataset (Sect. 3.6). The clustering threshold has a default value of the DOFS divided by the desired number of state vector elements (DOFS/$n$), but this can be configured by the user. The algorithm then performs a k-means clustering of proximate grid cell pairs ($p \approx 2$) for the remaining lower-information grid cells on the basis of latitude, longitude, and $a_{ii}$, and retains pairs in the state vector that exceed the clustering threshold for the sum of their estimated averaging kernel sensitivities. The procedure is repeated for $p \approx 3$, and so on. The iteration stops when either the desired number of state vector elements is reached, the maximum cluster size is reached, or if continuing iteration would assign greater than the desired number of clusters. In the latter case, the remainder of the state vector is filled with elements of the maximum cluster size to achieve the desired number of state vector elements. At that point the final state vector is specified.

Users can determine a suitable number of state vector elements to balance information content and computational cost by running the IMI preview for varying state vector sizes and comparing the estimated DOFS (Fig. 4). The IMI preview visualizes the clustered state vector and the gridded $a_{ii}$ of each state vector element for user inspection. By adjusting the clustering threshold and maximum cluster size with feedback from the IMI preview, users can effectively control the size distribution of state vector elements. For example, in the CONUS inversion (Sect. 4) the estimated information content over CONUS is bimodal, with many grid cells of relatively high information content and many with very low information content. This bimodal distribution causes too many grid cell clusters to be above the default clustering threshold (estimated DOFS/$n$), leading to a premature filling of the grid with background clusters of the maximum cluster size to achieve a 600-element state vector. If left unchecked, this would lead to low resolution background state vector elements having an outsized sensitivity to the observations. To fix this issue we apply a clustering threshold of 2.0. Alternatively, the user could increase the desired number of state vector elements, but the information content may not justify the added cost as illustrated in Fig. 4 with the DOFS asymptote.

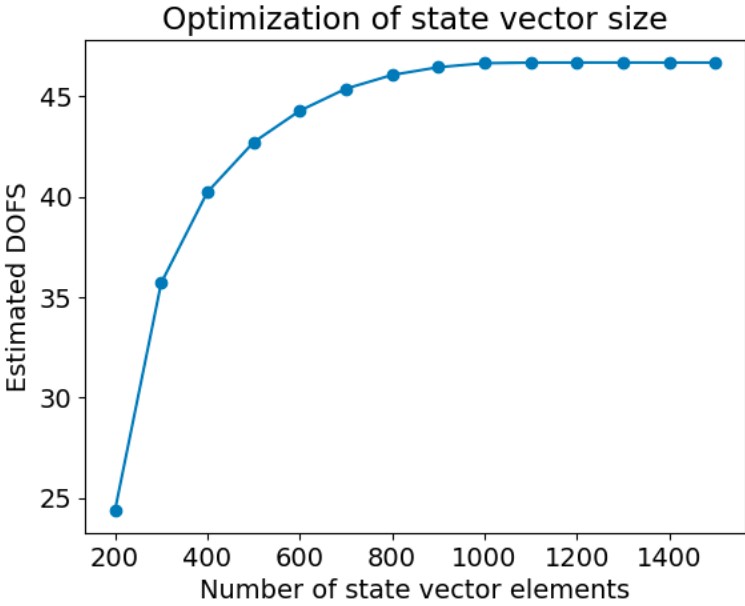


**Figure 4: Optimizing state vector size with the IMI preview (Sect. 2.2). The DOFS are estimated from the IMI preview for different user selections of state vector clustering sizes. Example is from the CONUS demo inversion (Sect. 4), where the gain in information content levels off beyond 800 elements. We use 600 elements for purpose of the demo.**

**3.6 Point source incorporation in the state vector construction**

We refer to point sources as emissions over 100 kg h$^{-1}$ from individual facilities. Point source satellite imagers with high spatial resolution are variably able to detect plumes from individual point sources and quantify emissions, but integrating this growing dataset into inversions is challenging due to uncertainties in point source observability, variability, and persistence (Cardoso-Saldaña and Allen, 2020; Cusworth et al., 2020; Watine-Guiu et al., 2023). Possible methods include enforcing
high state vector resolution where point sources have been observed (Chen et al., 2023), introducing additional point source state vector elements (Naus et al., 2023), and using point source data to evaluate posterior results in a TROPOMI-only inversion (Cusworth et al., 2020). Here, we integrate point source information into the IMI when constructing a reduced state vector (Sect. 3.5) by enforcing native resolution for state vector elements with point sources above a specified threshold (default of >2500 kg h$^{-1}$) for a minimum number of repeated observations (default of 50) in the continually updated global
datasets from SRON (Schuit et al., 2023; Methane Plume Maps, 2024), IMEO (International Methane Emissions Observatory, 2024), and Carbon Mapper (Cusworth et al., 2024). These datasets include satellite point source information from TROPOMI, Sentinel-2/3/5, EMIT, PRISMA, EnMAP, GOES, and Landsat-8/9. Point source locations are included as an overlay on the prior emission maps in the IMI preview.

### 3.7 Optimization of methane sink

IMI 1.0 only optimized methane emissions, but sinks also affect the methane concentrations. Global inversions commonly optimize tropospheric OH (the main methane sink) in the state vector as a methane loss frequency separately from emissions (Maasakkers et al., 2019; Yin et al., 2021). A common misconception is that regional inversions (either Eulerian or Lagrangian) do not need to optimize OH because the ventilation time scale is much shorter than the methane lifetime (Sheng et al., 2018), but the exact same consideration would apply to the effect of emissions. In IMI 2.0 we provide the option to

optimize the tropospheric OH concentration (as the methane loss frequency) averaged over the regional domain, or as hemispheric quantities for global inversions (Sect. 3.8), Prior estimates are the 4°×5° global 3-D monthly fields of OH concentrations used in GEOS-Chem (Wecht et al., 2014). These fields yield a global methane lifetime of 10.7 years against tropospheric OH, within the observationally constrained range of 11.2 ±1.3 years from proxy observations (Prather et al., 2012). Users can modify the default GEOS-Chem configuration with alternative OH prior estimates if desired. The default

prior error standard deviation for the inversion is 10% (Y. Zhang et al., 2018) but can be modified by the user in the configuration file. We find in our default regional inversions that optimization of tropospheric OH is effectively done by optimization of the BCs, rather than by optimization of OH itself, so that the methane loss frequency does not need to be in the state vector. This is because the default OH concentration field is relatively smooth. Users may want to include OH in their regional inversions if they replace the default OH concentration field or if they modify the default prior error variances

to have lower errors on BCs and/or higher errors on OH.

### 3.8 Global inversion capability

IMI 2.0 supports global inversions with optimization of both methane emissions at up to 2°×2.5° resolution and hemispheric OH concentrations (two additional state vector elements), following Qu et al. (2021). The same analytical inversion method used in regional inversions is applied globally. The global inversion is necessarily coarser than regional inversions. The

Kalman filter approach (Sect. 3.9) for time-dependent optimization of emissions can be used.

A critical requirement for global inversions is unbiased initial conditions. The smoothed TROPOMI archive described in Sect. 3.4 is effective for this purpose. It provides an unbiased synchronous representation of the stratosphere at the initial time that can then evolve with the forward simulation.

### 435 3.9 Temporal emission updates with Kalman filter

IMI 2.0 enables temporal emission updates and continuous emission monitoring at low latency with a Kalman filter approach as described by Varon et al. (2023). This workflow is depicted in Fig. 5. The IMI conducts successive methane inversions over user-specified time intervals (such as weekly or monthly), using observations for that interval. Prior emissions are taken

as the posterior emissions of the preceding interval (Kalman filter), the original bottom-up estimates, or a combination of the two through a nudge factor α (α= 0 for preceding interval, α = 1 for original bottom-up estimate). Relative error standard deviations on the prior emissions are not updated from their original values (50% in the default) because of unresolved prior error on the temporal variability of emissions. Varon et al. (2023) used α = 0.1 in their weekly inversions for the Permian Basin to retain some information from the original prior distribution and to prevent state vector elements from getting locked at low values when a fixed relative error on the prior emission is assumed. The state vector can be updated from one interval to the next with the adaptive k-means clustering scheme (Sect. 3.5) to take advantage of changing spatial patterns in information content. The latency of temporal emission updates is dependent on the availability of the IMI input data. At present the latency is about 1 month behind real time, limited by availability of the GEOS meteorological data to drive GEOS-Chem.

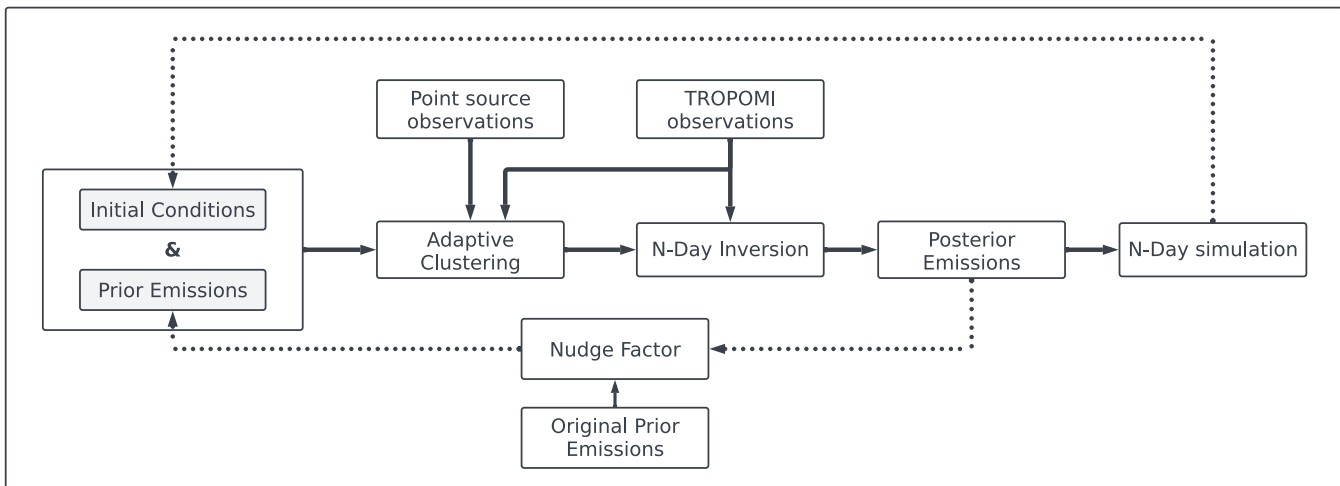

**Figure 5: Flowchart of the IMI 2.0 Kalman filter methodology for temporal emission updates with adaptive clustering enabled. Starting from initial conditions and bottom-up emission inventories used as prior estimates, the IMI reduces the state vector dimension by adaptive clustering (if desired) and conducts an inversion of TROPOMI observations for an N-day period to quantify the posterior emissions as averages for that period. At the end of the period, the posterior emissions are nudged back towards the original prior emissions (if desired) and used as prior estimates for the next N-day period.**

### 3.10 Updated bottom-up inventories for use as prior estimates

Table 3 lists the default bottom-up inventories used as prior estimates in IMI 2.0. Users may add or substitute their own prior estimates as NetCDF files to be read through HEMCO (Lin et al., 2021). New bottom-up inventories introduced in IMI 2.0 include worldwide emissions from hydroelectric reservoirs (Delwiche et al., 2022), an updated gridded version of US emissions from the EPA Greenhouse Gas Inventory (Maasakkers et al., 2023), and updated global anthropogenic emissions from EDGAR v8 (Crippa et al., 2023). Bottom-up anthropogenic inventories are reported with lag times of a few years or more and we use as default the most recent reported year, but this is of little consequence because most reported

anthropogenic emissions and their distributions generally vary little from year to year. Wetland emissions can have large year-to-year variability but also large uncertainties, so we use as default the monthly emissions from the mean of the WetCHARTs inventory ensemble for individual years (Bloom et al., 2021), which users can replace by monthly WetCHARTs or LPJ-wsl climatologies. GFED4 open fire emissions have daily temporal resolution and are for individual years.

**Table 3: Default prior emission inventories in IMI 2.0**

| **Anthropogenic** | |
| --- | --- |
| United States | EPA Gridded GHGI v2 – Express Extension (Maasakkers et al., 2023) |
| Mexico | INECC (Scarpelli et al., 2020) |
| Canada | ECCC NIR (Scarpelli et al., 2022a) |
| Rest of world | |
|    -   Fuel Exploitation | GFEI v2.0 (Scarpelli et al., 2022b) |
|    -   Other | EDGAR v8 (Crippa et al., 2023) |
| Hydroelectric reservoirs | ResME (Delwiche et al., 2022) |
| **Natural** | |
| Wetlands [a] | WetCHARTs v1.3.1 (Bloom et al., 2021); LPJ-wsl (Z. Zhang et al., 2016) |
| Geological seeps | Etiope et al. (2019) scaled to 2 Tg a$^{-1}$ (Hmiel et al., 2020) |
| Open fires | GFED4 (Randerson et al., 2017) |
| Termites | Fung et al. (1991) |

[a] The IMI uses as default monthly wetland emissions from the mean of the WetCHARTs ensemble, but users can replace this default with WetCHARTs or LPJ-wsl climatologies.

### 3.11 Lognormal error PDFs for heavy tailed emissions

IMI 2.0 includes an option to use lognormal error PDFs for the prior emission estimates. The frequency distribution of methane emissions often exhibits heavy tails that can be better characterized using lognormal errors rather than normal errors (Yuan et al., 2015; Cui et al., 2019). An added benefit of using lognormal errors is that it enforces positivity in the posterior solution (Miller et al., 2014), considering that we do not optimize the soil sink in the IMI. Following Maasakkers et al. (2019), we implement lognormal error PDFs for the prior emission estimates by optimizing for the logarithm of the emissions and otherwise using the same equations. The forward model relating methane concentrations to the logarithm of emissions is nonlinear, requiring an iterative approach to find the solution through repeated update of the Jacobian matrix **K** (Rodgers, 2000). Updates to **K** in log space are readily computed through simple scaling without having to re-run GEOS-Chem (Maasakkers et al., 2019). The iterative nature of the solution increases the computational cost dependent on the number of iterations needed to reach convergence. Only the prior emission elements of the state vector in the region of

interest are optimized in log space; buffer elements, boundary conditions, and OH concentrations continue to be optimized with normal errors (Maasakkers et al., 2019; Chen et al., 2022).

When using the lognormal error PDFs option, the inversion optimizes the median of the posterior PDF, starting from the median of the prior PDF. But the prior estimates reported in Table 3 should be viewed as the means of their PDFs, with a geometric error standard deviation $\sigma_g$ (for example, $\sigma_g = 2$ states a factor of 2 uncertainty). The median of a lognormal PDF is related to its mean by $\boldsymbol{x_{median}} = \boldsymbol{x_{mean}} \exp\left[-(\ln \sigma_g{}^2)/2\right]$ and we apply this correction to the prior estimates for input to the inversion. The inversion then returns the median of the posterior PDF with posterior error covariance matrix in log space. We convert the median to the mean of the posterior PDF with geometric error standard deviations by following the reverse of the above procedure. See Hancock et al. (2025) for further details on the method.

### 3.12 Enriched output

IMI 1.0 output included gridded posterior emission estimates with error standard deviations, averaging kernel sensitivities, and tables of emission totals. It also compared the GEOS-Chem simulations with posterior versus prior emissions to the TROPOMI observations as diagnostic of the improved fit resulting from the inversion, with means and spatial standard deviations of the time-averaged GEOS-Chem – TROPOMI differences as indicators. Here we add tabulated and gridded posterior emissions by source sectors, using the prior sectoral contributions within individual grid cells to assign corrections from the inversion to individual sectors (Wecht et al., 2014). This assumes that the relative contributions from different sectors within individual state vector elements are correct, which is a better assumption at high resolution because emissions within a given state vector element are then more likely to be dominated by a single sector. This also assumes that the prior error is the same for all sectors, which could be improved by using sector-dependent prior error variances (Shen et al., 2021). Sectoral output can be further partitioned by individual countries within the inversion domain. We also provide timeseries of emissions when using temporal emission updates (Sect. 3.9). Output for the IMI preview now includes gridded visualization of both TROPOMI datasets (operational and blended), overlay of point source locations (Sect. 3.6) on the map of prior emissions, and visualization of estimated averaging kernel sensitivities. The averaging kernel sensitivities are estimated in the preview using the density of super-observations (Sect 3.3) with a modified version of the IMI 1.0 formula in Eq. (4):

$$a_{ii} = \frac{\sigma_{a_i}^2}{\sigma_{a_i}^2 + \dfrac{(\sigma_{super,i}/k)^2}{m_{super,i}}}, \qquad (6)$$

where the variables are the same as in Eq. (4) except $\sigma_{super,i}$ and $m_{super,i}$. $\sigma_{super,i}$ is the observational error standard deviation for state vector element $i$ calculated as in Eq. (5) using the average number of grid cell observations, $P$, within that

element over the inversion time period. $m_{super,i}$ is the number of super-observations for state vector element $i$ over the inversion time period. The IMI configuration file is also written to the output directory for user reference.

### 3.13 IMI Docker container

IMI 2.0 includes a software container to facilitate installation of the IMI to local systems and support scheduled inversion workflows (e.g., once a week or once a month). A container is a lightweight, standalone, and executable software package that encapsulates an application and all its dependencies, including libraries, frameworks, and system tools. Creating an IMI container provides a stable and reproducible environment. It ensures that the IMI can run consistently across different systems, such as local clusters, cloud servers, and even local computers regardless of operating system. Once the container is

downloaded, the only dependency needed to run the software is Docker® (a container engine). The necessary input data is automatically downloaded from the AWS cloud upon running the IMI container.

The IMI container is built in two stages: a base container and an operational container. The base container builds the environment and dependencies needed to run the IMI (python packages, forward model dependencies, system tools). The

build of the base environment is performed with two commonly used scientific package managers, Spack™ and Micromamba™. The operational container build simply downloads and configures the IMI source code into the container. The operational container is built automatically via GitHub Actions upon new IMI version releases and archived on a publicly accessible cloud repository with download instructions on the IMI documentation site.

An application of the containerized IMI is to support the US GHG Center, a multi-agency initiative to provide a trusted repository of greenhouse gas data from models and observations on the AWS cloud (U.S. Greenhouse Gas Center, 2024). The US GHG Center uses a Multi-Mission Algorithm and Analysis Platform (MAAP; Earthdata, 2024) cloud environment designed to ingest containers. The IMI operates within MAAP to provide an inversion tool as part of the US GHG Center capabilities.

### 4 Example application to the contiguous United States


To demonstrate the new capabilities of the IMI, we show an example out-of-the-box application of IMI 2.0 to quantify methane emissions in the contiguous United States (CONUS). The inversion period runs from January 1, 2023 to December 31, 2023 with 28-day Kalman filter emission updates and the blended TROPOMI+GOSAT observations. The prior emissions are the defaults described in Sect. 3.10. The Kalman filter is designed to resolve any seasonal variation not

included or wrongly included in the prior estimates (Table 3). The CONUS domain is provided as a shapefile, and Canada and Mexico are included as 16 buffer clusters. The inversion uses GEOS-Chem as forward model at 0.25°×0.3125°

resolution which is therefore the native resolution of the state vector, corresponding to 11,698 elements over CONUS. We apply adaptive state vector clustering to reduce the state vector size from 11,698 to 600 elements for each 28-day inversion interval, as described in Sect. 3.9 and illustrated in Fig. 3. We apply a clustering threshold of 2.0 to prevent a bimodal size

distribution in elements with high information content, as described in Sect. 3.5. We apply a regularization parameter $\gamma = 0.2$ to prevent overfit. Other configuration settings for this demo CONUS inversion are given in Table 2.

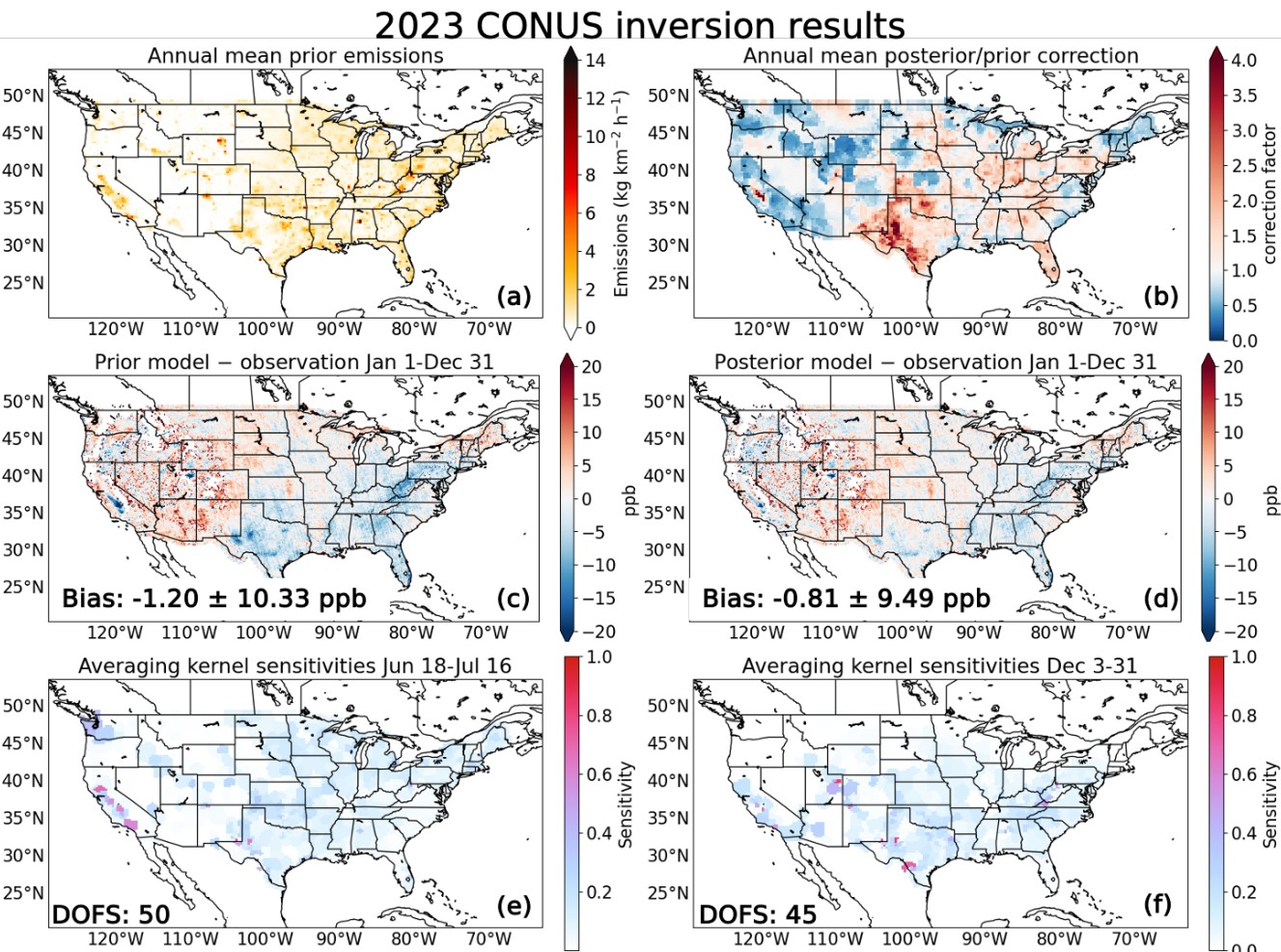

**Figure 6: Application of IMI 2.0 to a demo CONUS inversion with configuration settings given in Table 2. The**
**inversion is for 1 year (January 1 – December 31, 2023) over the CONUS domain provided as a shapefile and with buffer clusters for Canada and Mexico. Observations are from the blended TROPOMI+GOSAT product (Balasus et al., 2023). The top row shows the annual average prior emissions (a) and correction factors from the inversion (b). The middle row shows the differences in ppb between the observations and simulated concentrations using the original prior emissions (c) and the posterior emissions (d) for each inversion interval. Mean bias and spatial**
**standard deviations are given inset. The bottom row shows the averaging kernel sensitivities for the June 18 – July 16 (e) and December 3 – 30 (f) inversion intervals with DOFS inset.**

Figure 6 shows the spatial distribution of results as returned by the IMI. Starting from the EPA Gridded GHGI v2 – Express Extension taken as prior estimate on January 1, 2023, together with other prior estimates (Table 3), the inversion applies

mean correction factors for each 28-day period resulting in annual mean correction factors shown in Figure 6. A GEOS-Chem simulation with these posterior emission estimates provides a better fit to the TROPOMI data over the inversion domain than with the prior estimates, as indicated by the monthly mean bias and spatial standard deviation of the Δ(GEOS-Chem – TROPOMI) difference. The averaging kernel sensitivities returned by the inversion are highest, approaching unity, in regions of high emissions, and near zero in regions of low emissions. They are generally higher in summer than winter for

northern CONUS because of a higher fraction of successful TROPOMI retrievals.

Figure 7 shows the time series of CONUS 28-day emissions and the mean annual totals by sector. The annual mean prior emission is 34 Tg a$^{-1}$ and peaks in July. We find an annual mean posterior emission of 42 Tg a$^{-1}$ for 2023 with mean DOFS per 28-day inversion interval of 49 and maximum emission in September. The seasonal offset from July to September is

largely driven by wetlands, which may be explained by WetCHARTs' use of air temperature rather than soil temperature to predict wetlands emissions. The same issue is found for boreal wetlands (East et al., 2024). Upward corrections to emissions are largest for livestock and oil/gas, as is apparent in the posterior/prior correction patterns in Figure 6. Our results are broadly in agreement with the range of emission estimates for other CONUS inverse studies (Lu et al., 2022, 2023; Worden et al., 2022; Shen et al., 2023; Nesser et al., 2024a). Differences could be investigated in the IMI with sensitivity inversions

swapping prior emissions, observational products, and inversion parameters.

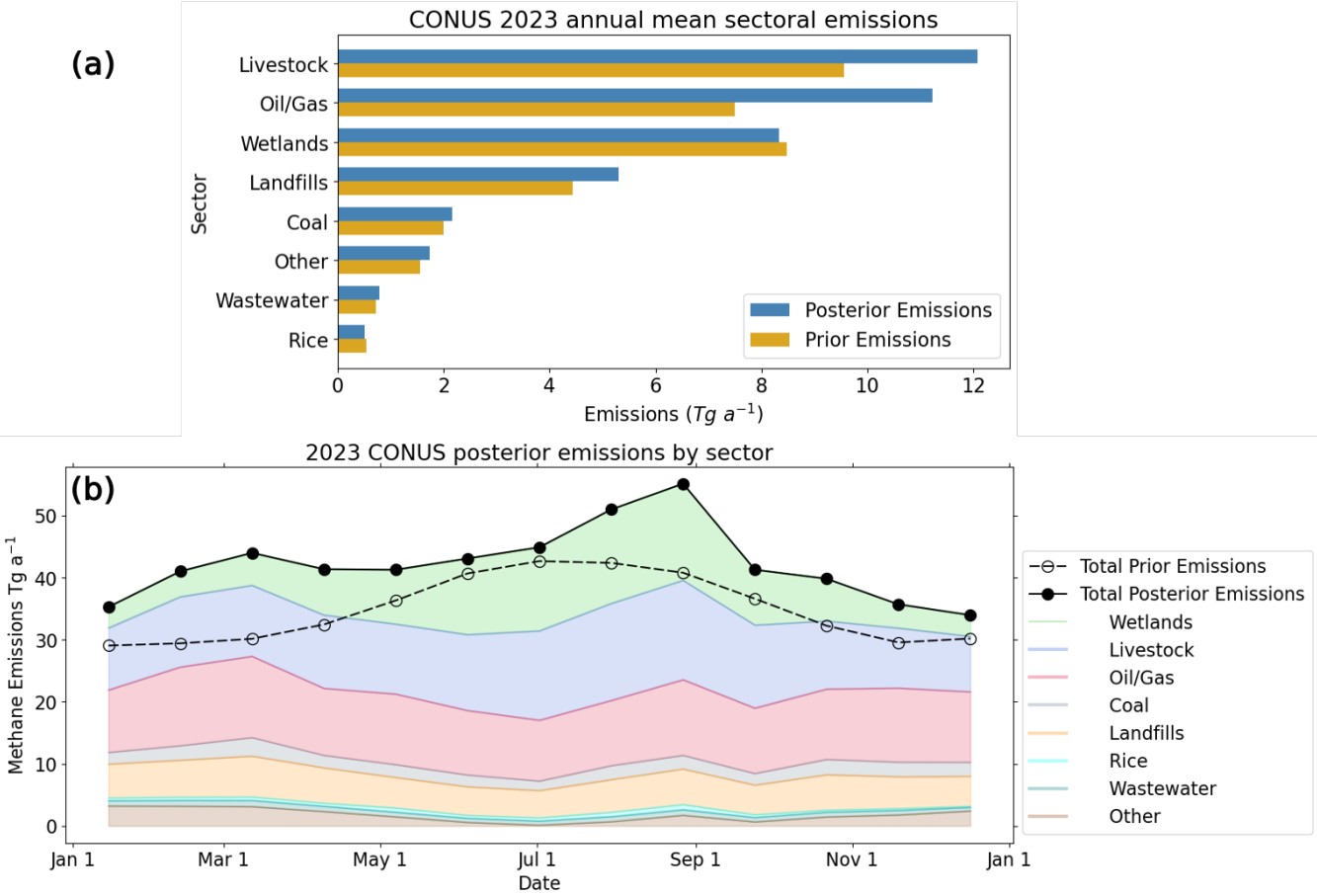

**Figure 7: CONUS sectoral emissions for 2023 and 28-day variability as obtained by the demo inversion using IMI 2.0. (a) Compares annual mean posterior and prior 2023 emission estimates by sector. (b) Shows the evolution of posterior sectoral emissions over 28-day inversion intervals, along with total posterior and prior emissions. Each symbol on the prior/posterior emissions line represents the middle of the 28-day inversion interval.**

## 5 Current limitations

While IMI 2.0 introduces major improvements over IMI 1.0 and can be regarded as state of the art for methane emission estimation using TROPOMI observations, there remain important limitations that will need to be addressed in future versions. Bias in the observations is a particular concern. Operational TROPOMI retrievals have been improving rapidly (Lorente et al., 2023), and machine learning based corrections have brought further improvements (Balasus et al., 2023; Bradley et al., 2024), but artifacts can still remain. The independent U. Bremen TROPOMI retrieval product (Schneising et al., 2023) will be important to include in future IMI versions.

Observations from surface sites and aircraft campaigns can provide independent evaluation of the posterior simulation and this is good practice in IMI applications (Varon et al., 2023; Hancock et al., 2025), but they could also be used as alternative or joint datasets in the inversion and this will be included in future IMI versions. Observations from the recently launched MethaneSAT satellite instrument will also be brought into the IMI when mature. Observations from the growing fleet of point source imagers should be more completely exploited, the main unresolved concern being the persistence of the

detected point sources (Cusworth et al., 2021).

The 25-km resolution of the IMI is presently limited by the resolution of the NASA GEOS meteorological product used to drive the GEOS-Chem forward model. A GEOS data set at 12-km resolution has recently been generated and is being applied in a prototype version of the IMI for 12-km urban inversions (Wang et al., 2024). Yet higher resolution together with

more precise transport can be achieved in the future with the high-performance GEOS-Chem (GCHP; Martin et al., 2022) and by using the Weather Research Forecasting (WRF) model to drive GEOS-Chem (WRF-GC; Feng et al., 2021).

IMI applications are presently limited to 1 month behind real time because of the workflow in processing the GEOS meteorological data to drive GEOS-Chem. We are working to lift this limitation, enabling inversions up to 2-3 days behind

real time. This will provide greater capability for near-term monitoring but the satellite observation density will then become limiting. Varon et al. (2023) found that weekly inversions of TROPOMI observations could be achieved over the Permian Basin but this is a favorable observing environment.

The ability to quantify emissions at high resolution is an underdetermined problem, because of the limitations in

observational coverage, so that prior inventory estimates play an important role in guiding inversion results. Errors in the prior distribution of emissions propagate to bias in the inversion results (Yu et al., 2021), and errors in the contributions from different sectors propagate to errors in sectoral attribution. We plan to continually update the prior emission inventory database in the IMI with improved products, such as GFEI v3 (Scarpelli et al., 2025) and the Global Rice Production Inventory (GRPI; Chen et al., 2025). Error characterization in the prior estimates is a difficult issue. Inversion results are

sensitive to the choice of prior error estimates and whether a normal or lognormal error PDF is assumed. Spatial error correlations in the prior estimate are also certainly present but difficult to define and have been ignored for now.

**6 Conclusions**

The Integrated Methane Inversion (IMI) is a cloud-based, user-friendly software tool for researchers and stakeholders to infer methane emissions from TROPOMI satellite observations. Here we presented major new developments in IMI 2.0 that

increase its value and reliability to quantify emissions from the global scale down to 25-km resolution, incorporating also additional information from point source imaging satellite instruments. IMI 2.0 features a capability for low-latency

monitoring of the temporal variability of emissions for any user-selected region. It can be used on the cloud or downloaded to local compute clusters. Full documentation and a user's manual are available at https://imi.readthedocs.io.

The IMI is a living software tool with a steady stream of development from scientific users. We have already launched new developments to include: (1) reducing latency to 2-3 days; (2) increasing the spatial resolution to $12\times12$ km$^2$; (3) integrating methane observations from new satellite instruments (MethaneSAT); (4) incorporating information from point source imagers (GHGSat, Carbon Mapper) directly into the inversion; (5) adding an IMI preview tool to provide guidance on boundary conditions; (6) extending the IMI to $CO_2$. These and other developments will provide the basis for IMI 3.0.


*Code availability*

The IMI source code and documentation is available at https://imi.seas.harvard.edu/. The code used in this paper is permanently archived at https://zenodo.org/doi/10.5281/zenodo.6081933.

*Data availability*

The TROPOMI methane data are available on the Amazon Web Services (AWS) cloud at https://registry.opendata.aws/sentinel5p/ (last access: 26 April 2024; AWS, 2024). The blended TROPOMI+GOSAT data are available at https://registry.opendata.aws/blended-tropomi-gosat-methane/ (Balasus et al., 2023). The GEOS-FP emission fields, boundary condition fields, and meteorological fields are available on AWS at https://registry.opendata.aws/geoschem-

input-data/ (last access: 25 April 2024; AWS, 2024).

*Author contributions*

LAE, DJV, MS, and DJJ contributed to the study design. LAE, DJV, MS, NB, SHE, MH, AOO, and JDE developed the model code. LAE, DJJ, DJV, NB, HN, and ZC contributed to the development of methods. LAE performed the analysis and

wrote the original draft. All authors reviewed and edited the manuscript.

*Competing interests*

The contact author has declared that none of the authors has any competing interests.

*Acknowledgments*

This work was funded by ExxonMobil Technology and Engineering Company, by the Harvard Initiative on Reducing Global Methane Emissions, the NASA Carbon Monitoring System, and by the NASA Jet Propulsion Laboratory initiative for the US GHG Center. Part of this research was carried out at the Jet Propulsion Laboratory, California Institute of Technology, under a contract with the National Aeronautics and Space Administration. This research was supported in part by an

appointment to the NASA Postdoctoral Program at the Jet Propulsion Laboratory, California Institute of Technology administered by Oak Ridge Associated Universities under contract with NASA.

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
