# Peer review of "Integrated Methane Inversion (IMI) 2.0: an improved research and stakeholder tool for monitoring total methane emissions with high resolution worldwide using TROPOMI satellite observations"

_EGUsphere, 2024_

## Author Response (AR1)

**Reviewer 1:**

*This study builds upon the Integrated Methane Inversion (IMI) v1.0 framework, advancing it to IMI 2.0 for monitoring total methane emissions with high resolution worldwide using TROPOMI satellite observations. The major advances involve the use of new observation data, improved consideration of inversion parameters (including state vectors and errors) and processes. These updates incorporate the outcome from a series of studies on methane inversion from the community. The capability of this new tool is demonstrated by a case demo of US methane emission. Overall, the manuscript is well-structured and well-written. It can be published once the following points are clarified.*

*Line 262: "we modified GEOS-Chem to simulate multiple methane species as separate transported tracers" It appears to me that the GEOS-Chem methane simulation only includes one tracer (methane), what does "multiple methane species" mean here? Are you referring to mean methane from different source?*

Thanks for pointing this out. The terminology is confusing. We have modified GEOS-Chem to be able to simulate multiple independent methane species such that methane emitted from one grid cell can be distinguished from methane that was emitted from another grid cell. We have updated the quoted text to be more clear.

*Line 272: "Precompiling emissions with HEMCO yields an additional 2× speed-up, for an overall 10× decrease in CPU cost and a net decrease in wall time. U" Could you explain what does "precompiling" refer to?*

Precompiling refers to aggregating the emissions from different sectors (and files) into a single authoritative file that can be read in at runtime to reduce Input/Output latencies. We have clarified this in the text and change it from precompile to precompute.

*Section 3.4. I am a bit confused about the term "super-observation". It looks to me that the number of observations is reduced, but "super" mispleads as "more and stronger".*

This is terminology introduced from Eskes et al. (2003). We have updated the text to reference this paper.

*Line 304. It mentions "three" updates but there are indeed four processes.*

Thank you for catching this error – updated!

*Line 335: I am curious about the extent to which boundary conditions would be adjusted in the inversion process if smoothed BCs were employed. Have the you tested this in the demo inversion?*

The boundary condition adjustment is typically by a few ppb, but may be more depending on the region of interest and the fidelity of the TROPOMI data. For example, over open oceans we apply the zonal average bias correction, so the inversion may apply a higher correction for domain edges over oceans. Indeed, for the demo inversion, the mean adjustment was ~20ppb on the east coast. The other domain edges had a mean adjustment of ~1ppb. We have added text in the boundary condition section to highlight the greater need for boundary condition optimization over oceans.

*Line399 : Could you provide a test demonstrating how OH optimization influences the regional inversion results? Why was OH optimization not utilized in the US demo?*

Thanks for raising this question. Information on OH typically gets overwhelmed by correction to the boundary conditions in regional inversions unless a strong spatial structure of the OH sink is imposed in the prior. The section on methane sink optimization has been updated to note this.

*Table 2: Could you provide a suggested values for the regularization parameter? Is it sensitive to the number of observations and state vector?*

Thanks for bringing this up. The regularization parameter is used to account for errors in the specification of the prior and observing system error covariance matrices. Usually, it is a value between 0 and 1. We have updated section 2 to include discussion of the regularization parameter.

**Reviewer 2:**
*Estrada et al. present an update to the Integrated Methane Inversion (IMI), a user-friendly tool to estimate methane emissions using TROPOMI XCH4 observations. The update to IMI 2.0 has extensive improvements over the original IMI, including the option to use a different TROPOMI product, different priors, point source satellite observations, improved BCs, lognormal PDFs, reduction of the state vector to improve computational efficiency and especially key, the use of Kalman filters to have near-real time continuous monitoring of emissions. The tool and its updates are valuable, and the paper is well written and structured. I recommend publication after the below comments are addressed.*
*These comments generally relate to ensuring newcomers are aware of some nuances that come with using atmospheric inversions. I suggest this could be done in a separate 'limitations' section, or the individual points can be discussed where they are relevant throughout the paper.*

1. *Near-real time inversions are an impressive and important forward step. However, it feels as though the paper needs a discussion of the issue of lack of validation data that comes with near-real time inversions. For example, anomalies and biases in*

*TROPOMI products have sometimes been found after they have already been used in analyses. Validation of the satellite inversion results could also be carried out with independent surface-based observations, so that users can see an extra level of validation for the years with which these observations are available. Please could a discussion on validation be included. Could this also be a future aim of the IMI? Additionally, it would be interesting to know if you are intending on including both surface and satellite observations within the inversion framework in the future.*

2. *The inclusion of another TROPOMI product (the GOSAT blended product) is valuable to compare inversion results from different products. However, I am surprised that there is no mention of the WFMD product from the University of Bremen. While it is not available in near-real time, it has a different retrieval algorithm from the operational product, making it useful to check for systematic biases. In some contexts (e.g. northern high latitudes) the operational product has been shown to have higher seasonal biases than the WFMD product (e.g. Lindqvist et al., 2024 - https://www.mdpi.com/2072-4292/16/16/2979).*

3. *IMI 2.0 currently relies on a single transport model and inversion framework, although some inversion parameters can be adjusted. I believe it would be helpful to acknowledge that both the choice of transport model and inversion framework can impact posterior emissions estimates. This would be particularly useful for new users who may not be aware of this potential impact.*

4. *If I am correct the inversion does not optimise separate sectors, and the emissions from each source sector are found using the prior fraction of emissions from each sector in each grid cell (line 483). Some inversion frameworks do optimise different sectors within the inversion (including a number in the latest Global Methane Budget - https://essd.copernicus.org/preprints/essd-2024-115/) . Please could a discussion on this be included.*

These are all great points! We have added a current limitations section (Section 5) to address the majority of these points. For point 4, we have added discussion in section 3.12 to bring up alternate methods for optimizing different sectors.

**Specific comments:**

*Line 43: Bottom-up methods also include process-based models (e.g. for wetlands).*

Good point, the text has been updated to reflect this.

*Line 71: The link https://imi.seas.harvard.edu/ did not work for me at the time of opening it (404 site not found).*

Thanks for catching this. We have updated the site domain to https://carboninversion.com/. I have updated the text to use the updated domain. We have also fixed https://imi.seas.harvard.edu/ to forward to the new domain as well.

*Line 79: Recently Vara-Vela et al., 2024 was published, which I believe also used the IMI. https://www.mdpi.com/2072-4292/16/23/4554*

Thanks, we have added them to the citation!

*Line 114: Please could you comment on estimating offshore emissions when TROPOMI retrievals are removed from the ocean?*

Offshore emissions can be calculated as long as the over land observations are sensitive to those emissions (through transport). We have added a sentence in the blended TROPOMI+GOSAT section that explains this.

*Line 161: This is a minor point and up to your discretion, but as someone more familiar with averaging kernel sensitivities in the satellite retrieval context, I initially found this section a bit confusing. Maybe it would be useful to make clear the distinction between a in satellite retrievals and a in this context?*

Good point, we have added text specifying that it is the averaging kernel for the inversion.

*Line 246: I think it would be useful here to briefly explain the reduced state vector alongside referring to Section 3.6. Without that context, it's unclear why you need to enforce native resolution at the point sources. Also, what is the specified threshold? Is it the 100kg h-1 referred to earlier?*

This is a good point. I have moved the point source section (now Sect. 3.6) to directly follow the clustering section (now Sect. 3.5) for clarity. The default threshold is >2500 kg h$^{-1}$( added to the text), but this is configurable.

*Line 315: Please could you explain what kriged means?*

Replaced kriged with interpolated.

*Section 3.7: Relating to Section 3.10, is it also possible for the user to define a different OH field? This is another thing that can have a big impact on posterior emissions.*

This is an excellent point. Users can modify the GEOS-Chem simulation to use other OH fields by direct modification of GEOS-Chem files. I have updated the text in section 3.7 to mention this.

*Line 397: Only the case for regional inversions using an Eulerian CTM?*

Strong spatial patterns in OH would cause biases in the emissions during the inversion for both Eulerian and Lagrangian models if not taken into account by the model. We have updated the text to note this is an issue for both models.

*Line 463: What is the impact on the natural soil sink of using a lognormal distribution that enforces positivity?*

We do not optimize the natural soil sink as part of the inversion. This prevents non-physical uptake that can occur when optimizing sinks. We have updated the text in section 3.11 to note this.